

# Urban heat forecasting in small cities: Evaluation of a high-resolution operational numerical weather prediction model

Yuqi Huang[1], Chenghao Wang[1,2], Tyler Danzig[3], Temple R. Lee[4], Sandip Pal[3]

[1]School of Meteorology, University of Oklahoma, Norman, 73072, USA
[2]Department of Geography and Sustainability, University of Oklahoma, Norman, 73019, USA
[3]Department of Geosciences, Texas Tech University, Lubbock, 79409, USA
[4]NOAA/Air Resources Laboratory, Oak Ridge, 37830, USA

*Correspondence to*: Chenghao Wang (chenghao.wang@ou.edu)

**Abstract.** With rising global temperatures, urban environments are increasingly vulnerable to heat stress, often exacerbated
by the Urban Heat Island (UHI) effect. While most UHI research has focused on large metropolitan areas around the world, relatively smaller-sized cities (with a population 100,000-300,000) remain understudied despite their growing exposure to extreme heat and meteorological significance. In particular, urban heat advection (UHA), the transport of heat by mean winds, remains a key but underexplored mechanism in most modelling frameworks. High-resolution numerical weather prediction (NWP) models are essential tools for simulating urban hydrometeorological conditions, yet most prior evaluations have
focused on retrospective reanalysis products rather than forecasts. In this study, we assess the performance of a widely used operational weather forecast model—the High-Resolution Rapid Refresh (HRRR)—as a representative example of current NWP systems. We investigate its ability to predict spatial and temporal patterns of urban heat and UHA within and around Lubbock, Texas, a small-sized city located in a semi-arid environment in the southwestern U.S. Using data collected between 1 September 2023, and 31 August 2024 from the Urban Heat Island Experiment in Lubbock, Texas (U-HEAT) network and
five West Texas Mesonet stations, we compare 18-h forecasts against in situ observations. HRRR forecasts exhibit a consistent nighttime cold bias at both urban and rural sites, a daytime warm bias at rural locations, and a pervasive dry bias across all seasons. The model also systematically overestimates near-surface wind speeds, further limiting its ability to accurately predict UHA. Although HRRR captures the expected slower nocturnal cooling in urban areas, it does not well capture advective heat transport under most wind regimes. Our findings reveal both systematic biases and urban representation limitations in current
high-resolution NWP forecasts. Our forecast–observation comparisons underscore the need for improved urban parameterizations and evaluation frameworks focused on forecast skill, with important implications for heat-risk warning systems and forecasting in small and mid-sized cities.



## 1 Introduction

Global cities are experiencing increasingly pronounced environmental changes driven by rapid urbanization, climate change, and other anthropogenic influences (Masson et al., 2020; Oke et al., 2017). In turn, cities have been shown to substantially alter local weather and climate processes, such as cloud and precipitation patterns (Lu et al., 2024; Vo et al., 2023; Yang et al., 2024), boundary layer development (Caicedo et al., 2019; Fenner et al., 2024; Pal et al., 2012), and air pollutant transport (Klein et al., 2014; Lac et al., 2013; Wang et al., 2018a). Among these, urban thermal conditions stand out as being particularly

affected, shaped by the heterogeneous nature of the built environment and surrounding semi-urban and rural areas. Indeed, heat-related phenomena, such as the urban heat island (UHI) effect, urban dry/moisture islands, and heat waves, have remained central focuses in urban climate research (Barlow, 2014)

Like other urban climatic phenomena, urban thermal conditions are strongly influenced by both atmospheric processes and surface properties (Oke et al., 2017). Wind plays a particularly important role by transporting heat via advection and turbulent

mixing, mechanisms that drive temperature gradients within and around urban areas, a phenomenon known as urban heat advection (Lowry, 1977). Both the intensity and spatial structure of the UHI are more closely linked to wind speed and direction compared to humidity or urban morphology (Bassett et al., 2019). Strong winds typically reduce UHI intensity by enhancing heat dispersion, whereas calm wind conditions tend to promote a circular UHI footprint (Oke, 1976). Under high wind conditions, the UHI often becomes elongated in the downwind direction due to advective heat transport (Bohnenstengel et al.,

2011). Beyond thermal effect, the dynamic urban heat advection process also influences local pollution dispersion and regional air quality (Agarwal and Tandon, 2010), and may affect convective processes such as cloud formation and precipitation patterns around cities (Wang et al., 2018b). A comprehensive understanding of the spatial variability in urban heat and its advection behavior is therefore critical for improving urban weather forecasting and mitigating temperature-related risks.

Understanding urban heat-related processes has traditionally relied on observations (Bassett et al., 2016, 2017). While such

observations offer high accuracy and real-time insights, they are inherently limited by sparse spatial coverage, lack of temporal continuity, and the high costs associated with installation and maintenance (Chen et al., 2022). Satellite remote sensing, by contrast, provides broad spatial coverage and effectively captures key urban surface characteristics (Zhou et al., 2018). However, it remains insufficient for characterizing urban thermal conditions, particularly due to its limited temporal resolution and the inability to retrieve key atmospheric variables such as air temperature and wind. In this context, Numerical Weather

Prediction (NWP) models have emerged as valuable tools for investigating urban heat processes, which provide full atmospheric properties at high temporal frequency with operational relevance (Best, 2005; Chen et al., 2011). Importantly, NWP forecasts, rather than reanalyses, are particularly critical for supporting urban resilience and heat mitigation through early warning systems (Kacker et al., 2025; Yang et al., 2016). Nevertheless, earlier land surface models (LSMs) commonly used in NWP systems, such as Noah-LSM (Chen and Dudhia, 2001), the Common Land Model (CLM) (Dai et al., 2003), and

Rapid Update Cycle (RUC) (Benjamin et al., 2004), often underrepresented urban processes due to coarse spatial resolution and limited urban-specific parameterizations, in part reflecting the operational emphasis on rapid forecasts.



Recent advances in high-resolution urban land-use datasets and urban land surface models (ULSMs) have substantially improved urban representation in numerical models (Chen et al., 2011; Lipson et al., 2024; Stewart et al., 2014). ULSMs, particularly those coupled with advanced Urban Canopy Models (UCMs) within frameworks such as WRF-SLUCM, now

offer a range of structural complexities from simple one-dimensional slab models to multilayer schemes that explicitly represent street canyons, walls, and road surfaces (Garuma, 2018). Among these, slab models remain widely used in operational NWP systems due to their computational efficiency (Oleson et al., 2008) and reasonable skill in simulating urban surface energy fluxes over predominantly impervious surfaces with strong sensible heat fluxes (Jongen et al., 2024; Lipson et al., 2024). However, these models idealize the urban surface as a homogeneous layer and oversimplify radiative and

hydrological interactions that are increasingly important in cities with nature-based solutions such as urban vegetation, green infrastructure, and irrigation. In these cases, more sophisticated UCMs have been shown to produce more realistic simulations of surface energy balance and near-surface meteorological conditions (Lipson et al., 2024). While slab models have been critiqued for their structural simplicity and are often considered surpassed by advanced UCMs in research applications, they remain the default choice in many operational forecasting systems. Despite this continued use, there has been limited evaluation

of how well these models perform in forecasts, especially with respect to capturing fine-scale spatiotemporal variations in urban heat. Addressing this gap is crucial for enhancing urban heat forecasting and supporting the development of more accurate and adaptive urban land surface parameterizations for operational use.

In addition, urban climate studies have predominantly focused on major metropolitan areas, often overlooking small to mid-sized cities due to limitations such as low model resolution and sparse observational networks. Yet, there are numerous small

cities around the world that remain understudied in urban climate research. For example, approximately 96% of U.S. cities with populations over 20,000 have fewer than 300,000 residents (U.S. Census Bureau, Population Division, 2024). Despite their smaller size and lower population density, these cities are increasingly vulnerable to heat stress due to limited adaptive capacity, constrained resources, and inadequate emergency response systems (Youngquist et al., 2023). Evidence from cities such as Szeged, Hungary, and Utrecht, the Netherlands, demonstrates that smaller urban areas can exhibit substantial UHI

intensities, emphasizing their susceptibility to elevated temperatures and associated risks (Brandsma and Wolters, 2012; Unger et al., 2011). This vulnerability is further amplified in dryland regions considering their fragile ecosystems and chronic water scarcity (Huang et al., 2017; Lee et al., 2023a, 2025). In these areas, decreasing soil moisture and vegetation cover contribute to a positive feedback loop of surface warming, land degradation, and further aridification (Charney et al., 1977; Li et al., 2021; Zhang et al., 2020), which may in turn exacerbate UHI effects. As population growth and climate fragility increase in

semi-arid regions, small and mid-sized cities are becoming critical frontiers for addressing heat-related challenges.

In this study, we conduct a year-long evaluation of urban heat forecasts from the High-Resolution Rapid Refresh (HRRR) model in Lubbock, Texas, a small city located in a semi-arid region of the southwestern U.S. HRRR is one of the highest-resolution operational NWP systems and incorporates a slab-type urban canopy parameterization. We use observational data collected from 23 Urban Heat Island Experiment in Lubbock, Texas (U-HEAT) stations measuring 2-m air temperature and

dew point temperature. Wind speed and direction data from five West Texas Mesonet stations are also used to characterize





local advection. We focus on three key aspects of urban thermal conditions: near-surface hydrometeorological forecasts, the spatial variability of UHA, and nocturnal cooling rates. While HRRR has been applied in urban climate studies, such as urban air quality assessments (Nauth et al., 2023), urban flood forecasting through coupled mesoscale models (Coelho et al., 2022), and urban boundary layer analysis (Strobach et al., 2024), its ability to simulate urban thermal conditions has not, to our

knowledge, been systematically evaluated. To address this critical gap, we pose the following research questions: (1) How accurately does HRRR simulate near-surface hydrometeorological conditions, particularly temperature and humidity, within and around a semi-arid urban environment across different times of the day and throughout varying seasons? (2) To what extent does HRRR reproduce observed nocturnal cooling rates across heterogeneous urban and rural landscapes? (3) Can HRRR capture the spatial variability and magnitude of UHI and UHA, particularly under varying wind regimes? By explicitly

addressing these questions, our findings aim not only to systematically evaluate HRRR's skill in urban heat forecasting across critical temporal scales but also to directly inform enhancements in operational NWP systems. Ultimately, this research seeks to improve forecast-based heat-risk management strategies for small and mid-sized cities facing growing heat vulnerability.

## 2 Datasets and methods

### 2.1 Study area

The study area, Lubbock, Texas (33.58°N, 101.84°W), is situated in the northwestern part of Texas within the Great Plains region (Fig. 1). As of the 2023 census, Lubbock had a population of 266,878, considerably smaller than major U.S. metropolitan areas such as New York City, Los Angeles, and Chicago. Despite recent population growth, Lubbock remains a medium-sized city. Its rapid urbanization, semi-arid climate, and relative isolation from the influence of larger metropolitan areas makes it an ideal testbed for examining thermal conditions in small, climate-sensitive cities.

According to the Köppen climate classification (Cui et al., 2021), Lubbock exhibits a cold semi-arid climate, characterized by an average annual precipitation of approximately 466 mm and a mean annual temperature of 16.3°C. The majority of precipitation occurs between May and October. While the overall decadal mean precipitation has shown a declining trend in recent decades, May remains an exception, exhibiting relatively stable or even increased precipitation levels (Fig. S1b). Concurrently, both monthly and decadal mean temperatures have demonstrated a consistent upward trend, with the decadal

mean temperature rising from 15.49°C to 17.21°C over recent decades (Fig. S1a). In the context of ongoing climate change, which is amplifying the frequency and intensity of heatwaves (Meehl and Tebaldi, 2004), Lubbock has experienced a notable increase in extreme heat events. Specifically, during the 2010s, the city recorded up to 171 days per year with maximum temperatures exceeding 100°F (37.8°C), indicating a significant escalation in heat stress conditions (Fig. S1a).





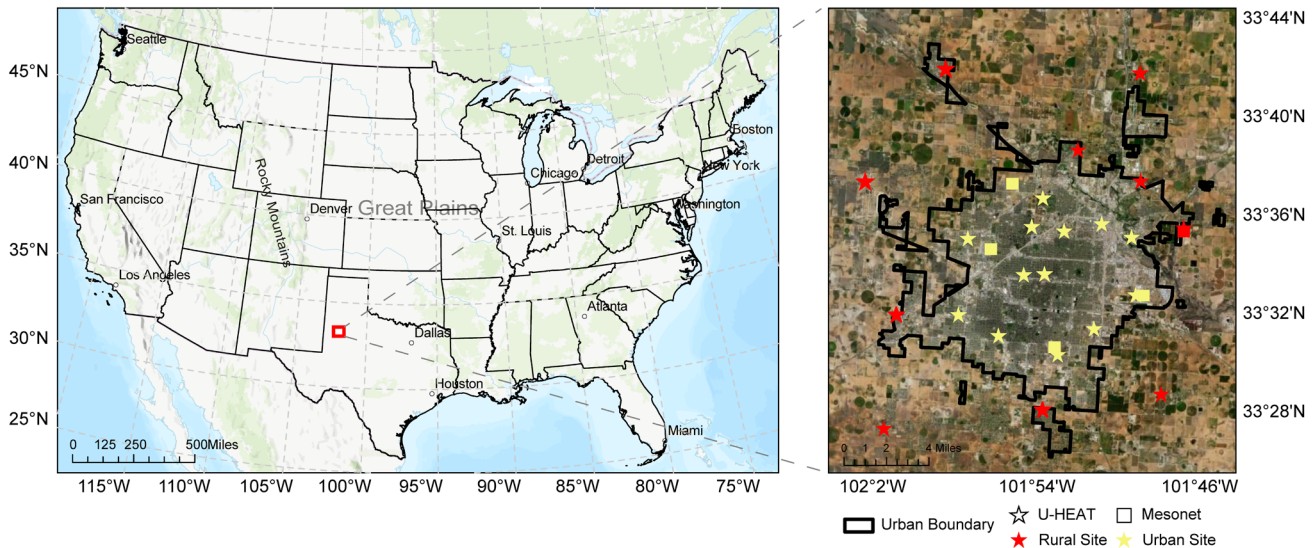

**Figure 1: Geographical overview of the study area: (a) location of Lubbock, Texas, within the contiguous United States (CONUS) shown on the topographic map; (b) distribution of 28 in-situ observational sites in and around Lubbock, including 23 U-HEAT stations and 5 West Texas Mesonet stations overlaid on a satellite image (map source: Esri, U.S. Department of Agriculture Farm Service Agency; urban boundary source: U.S. Census Bureau, 2020 Census).**

## 2.2 In-situ data

### 2.2.1 Observation networks in Lubbock

Accurately characterizing urban thermal conditions requires observational datasets that offer both high temporal resolution and extensive spatial coverage across urban and surrounding rural areas. To address this need, we leveraged observational data from two complementary networks: the West Texas Mesonet (WTM) and the newly established Urban Heat Experiment Around Lubbock, Texas (U-HEAT) network.

The WTM, established in 1999 through funding from the Texas Department of Economic Development, is operated by the National Wind Institute. For this study, we utilized one-minute interval observations of 10-meter wind speed and direction. Among the 158 active WTM stations, five located in and around Lubbock were selected due to their proximity to the urban core and their representative spatial coverage (Fig. 1b). These stations are Lubbock TTU East Campus (33.59°N, 101.78°W), Lubbock All Saints School (33.50°N, 101.88°W), Lubbock South Plains Food Bank (33.54°N, 101.81°W), Lubbock TTU (33.60°N, 101.89°W), and Lubbock LCU (33.57°N, 101.94°W).

The U-HEAT stations use HOBO MX2301A data loggers. Each logger is equipped with internal sensors for temperature and relative humidity, enabling the calculation of dew point temperature. The entire U-HEAT was deployed between June and August 2023 and began recording observations immediately upon installation. To capture detailed temperature variability during periods of extreme heat in summer and early fall, the loggers were configured with a 1-minute recording interval. For the remaining months, when UHI intensity is typically lower (Oke et al., 2017), a 5-minute interval was applied. A total of 23



stations were strategically placed across Lubbock (Fig. 1b and Table S1), aligned along the city's southwest-to-northeast urban development axis. This orientation, forming a round-shaped network, has been demonstrated in previous studies to enhance the accuracy of UHI intensity and UHA assessments (Bassett et al., 2016; Danzig, 2024). Sensor placement, including height, land-surface characteristics, and deployment practices, adhered to World Meteorological Organization guidelines, ensuring
installation on grassy surfaces, free from overhead vegetation and nearby anthropogenic heat sources.

### 2.2.2 Data pre-processing

We retrieved both the 18-h forecast and the 0-h forecast (i.e., the model initialization) from HRRRv4 for the period between 1 September 2023 and 31 August 2024, via the NOAA Open Data Dissemination (NODD) program through the Google Cloud
Platform HRRR archive. For site-specific analysis, the nearest HRRR grid point was identified for each observation location, resulting in a total of 24 HRRR grid points selected for direct comparison with in situ measurements (Fig. S2). To ensure temporal consistency, all U-HEAT observational data were converted to Coordinated Universal Time (UTC), while the Mesonet data were already recorded in UTC. We also took into account daylight-saving time in Lubbock, where the local time offset from UTC shifts from –5 hours to –6 hours between November 5, 2023, at 01:00 and March 10, 2024, at 02:00 due to
the transition to standard time. For the 1-minute interval data from the Mesonet and U-HEAT, hourly means were calculated by averaging observations within a 10-minute window cantered on each exact hour (i.e., 5 minutes before and after). This approach helps smooth short-term variability and reduce potential errors associated with localized fluctuations, such as those induced by transient urban influences or microscale atmospheric perturbations. For the U-HEAT data recorded at 5-minute intervals, the observation at the exact hour was directly used to represent the hourly value. A summary of the extracted HRRR
variables and corresponding observational datasets used for comparison is provided in Table 1.

**Table 2: Summary of variables retrieved from the observational networks and HRRR model**

| Data Source | Variables | Measurement height | Time |
|---|---|---|---|
| U-HEAT (23 sites) | Air temperature (°C) | 2 m | Local Time (CDT) |
| | Dew point temperature (°C) | 2 m | CDT |
| | Wind speed (m s$^{-1}$) | 10 m | Coordinated Universal Time (UTC) |
| | Wind direction (-) | 10m | UTC |
| HRRR (24 grids) | Air temperature (K) | 2 m | |
| | Dew point temperature (K) | 2 m | |
| | Westward wind component (m s$^{-1}$) | 10 m | UTC |
| | Southward wind component (m s$^{-1}$) | 10 m | |
| | Total cloud cover (%) | Entire atmosphere | |





### 2.3 HRRR forecast data

The HRRR model is a deterministic, convection-allowing numerical weather prediction system that provides high-resolution (3 km) short-term (hourly) forecasts across the conterminous United States (CONUS). Operational since 2014, HRRR has been widely used to support real-time weather forecasting (Benjamin et al., 2016; Lee et al., 2024). Our evaluation focuses on version 4 of HRRR (HRRRv4), which incorporates the Rapid Update Cycle Land Surface Model (RUC LSM) to simulate surface and near-surface hydrometeorological conditions. The RUC LSM includes 21 land-cover categories based on MODIS

classifications; however, it represents all urban areas using a single category (Category 13: Urban and Built-Up), which limits its ability to capture the heterogeneity of urban surface types (Smirnova et al., 2016). We specifically assess HRRR's 18-h forecast, as it represents the maximum lead time available in its hourly forecast cycle. This forecast horizon is also less influenced by initial conditions and data assimilation compared to shorter lead times, making it more suitable for evaluating the model's forecast capability, particularly its treatment of land–atmosphere interactions (Lee et al., 2024). Detailed

descriptions of the HRRR system and the RUC LSM can be found in (Benjamin et al., 2016) and (Dowell et al., 2022)

### 2.4 Model evaluation metrics

    The overall performance of HRRR was determined by thoroughly evaluating near-surface meteorological variables (2-m air temperature, 2-m dew point temperature, and 10-m wind speed) in different seasons and during daytime or nighttime against observational data. Here, daytime is defined as 12:00 to 05:00 pm local time and nighttime as 12:00 to 05:00 am local time,

efficiently capturing the periods of peak solar influence and nocturnal cooling. Evaluation metrics include mean bias (MBE), root mean square error (RMSE), and Pearson correlation coefficient ($r$), which are expressed as (Wang and Wang, 2020; Wilks, 2011):

$$MBE = \frac{1}{N} \sum_{i=1}^{N}(M_i - O_i) \tag{1}$$

$$RMSE = \sqrt{\frac{1}{N}\sum_{i=1}^{N}(M_i - O_i)^2} \tag{2}$$

$$r = \frac{\sum_{i=1}^{N}(M_i - \overline{M_i})(O_i - \overline{O_i})}{\sqrt{\sum_{i=1}^{N}(M_i - \overline{M_i})^2}\sqrt{\sum_{i=1}^{N}(O_i - \overline{O_i})^2}} \tag{3}$$

where $M_i$ and $O_i$ represent the model forecast and observation, respectively, an overbar (i.e., $\overline{M_i}$ and $\overline{O_i}$) indicates the mean of all samples, and $N$ denotes the number of simulated periods, which is 8,784 for a year-long simulation. In general, model performance is considered more accurate when the MBE and RMSE are close to 0, and $r$ approaches 1.





## 2.5 Determination of nocturnal cooling rate

The nocturnal cooling rate indicates how effectively an area losses heat accumulated during the day. Urban areas, characterized by high fraction of impervious surfaces, complex geometries, and distinct thermal properties, typically exhibit slower cooling rates than rural surroundings, leading to pronounced nighttime UHI effects (Oke et al., 2017). Several seminal studies have demonstrated that nighttime cooling is strongly modulated by surface properties, atmospheric conditions, and land cover patterns, making it a key metric for assessing nighttime thermal behavior and urban hydrometeorological conditions (Kidder and Essenwanger, 1995; Spronken-Smith and Oke, 1999). This contrast is particularly pronounced in semi-arid cities such as Lubbock, where prevailing clear skies and dry air conditions can enhance radiative cooling in rural areas and intensify urban–rural cooling differences. Here we consistently define nighttime as from 12:00 am to 05:00 am local time, aligning with the evaluation window used in model evaluation. To isolate the effects of surface characteristics and minimize cloud-induced variability, we restrict our analysis to nights with continuous domain-wide cloud cover below 25%. Only regression-derived cooling rates with statistical significance ($p < 0.05$) are retained for further analysis.

## 2.6 Assessment of urban heat advection

The general framework of estimating urban effects on local and regional climate was initially established by Lowry (1977). Although urban heat advection has been recognized in numerous UHI studies, it remains relatively underexplored, primarily due to the scarcity of dense observational networks and challenges in effectively isolating urban influences from those of the surrounding landscape or background climate (Lowry, 1977). Using measurements from only a few sites, Angevine et al. (2003) provided some qualitative ideas about the presence of UHA in their work. Building upon this foundational concept, Heaviside et al. (2015) introduced a novel approach to isolate UHA from the background UHI by decomposing UHI intensity into two components: (1) a time-mean temperature field representing background conditions aggregated over all wind directions, and (2) a wind-dependent temperature field driven by the prevailing flow. This method has been applied and validated in various studies utilizing both observational networks and numerical models (Bassett et al., 2016, 2019). However, to the best of our knowledge, weather forecast model has never been examined in the context of UHA.

In this study, we adopt a similar wind-direction-dependent analytical framework to quantify UHA forecasts from the HRRR model. Specifically, we use wind-direction-dependent rural reference temperatures, $T_{rural}$, following the procedural steps outlined in (Heaviside et al., 2015), but extend the method by refining the wind classification. Instead of the original four categories, we classify the mean nighttime 10-m wind field from HRRR into eight directional quadrants. To minimize noise from short-term directional shifts, only nights with relatively stable wind conditions (defined as having consistent wind direction over six consecutive nighttime hours) are included in the analysis. These filtered nights are referred to as "effective nights". For each effective night $t$, we derive the temperature anomaly $\Delta T_{(t)}$ at each grid cell as the difference between the




HRRR-forecasted temperature and the corresponding rural reference temperature $T_{rural}$. We then determine the annual mean temperature anomaly $\overline{\Delta T}_{annual}$ across all effective nights. Lastly, the wind-dependent UHA signal, denoted $UHA_{(annual)}^{(wind)}$, is derived by subtracting $\overline{\Delta T}_{annual}$ from the wind-direction-specific temperature anomaly. We calculated temperature anomaly

following a similar procedure using station-based wind direction data to indicate the urban heat advection inferred from observational data. Temperature anomalies at each site are computed during each effective night and then averaged across all effective nights for each wind direction. This approach isolates persistent spatial patterns in nocturnal temperature associated with mean wind advection. Since wind transports heat from warmer to cooler regions, a systematic downwind warming captured through these directional composites would serve as a robust proxy for UHA. The workflow for calculating HRRR-

derived UHA is illustrated in Fig. 2.

**Figure 2: Workflow for calculating Urban Heat Advection (UHA), adapted from the framework by (Heaviside et al., 2015).**





## 3 Results

In this section, we present a comprehensive evaluation of HRRR forecast performance against U-HEAT observations. In subsection 3.1, we assess overall forecast skill for three key urban hydrometeorological variables: 2-m air temperature, 2-m dew point temperature, and 10-m wind speed. Subsection 3.2 compares predicted hourly outputs at two lead times, i.e., 0-h (near-real-time) and 18-h, to assess sensitivity to forecast horizon. The final two subsections focus on HRRR's capability to capture the temporal and spatial features of urban thermal environments. Specifically, subsection 3.3 evaluates forecasted nocturnal cooling rates at urban and rural sites, while subsection 3.4 assesses HRRR-derived UHA in comparison with observation-based estimates.

### 3.1 Spatial, seasonal, and diurnal evaluation of HRRR hydrometeorological forecast

The comparison between HRRR 18-h forecasts and observations across the U-HEAT suggests generally strong correlations for both 2-m air temperature and dew point temperature throughout the day during the entire study period, with average correlation coefficients ($r$) across all sites of $0.98 \pm 0.00$ and $0.93 \pm 0.01$, respectively (Fig. 3). The average RMSE for dew point temperature ($4.06 \pm 0.19°C$) is notably higher than for 2-m air temperature ($2.01 \pm 0.12°C$). There is no clear urban–rural contrast in RMSE, though slightly better performance is observed in the southwestern part of the study domain for 2-m air temperature. In terms of MBE, urban sites exhibit a consistent cold bias for 2-m air temperature (average MBE = $-0.27°C$), whereas peripheral rural sites show a warm bias (average MBE = $0.10°C$). In comparison, HRRR forecasts uniformly underestimate dew point temperature at both urban and rural sites, with an average MBE of $-1.95 \pm 0.22°C$. HRRR struggles to accurately capture wind speed across all five Mesonet sites, irrespective of their urban or rural classification, as evidenced by low correlations ($r$ ranging from 0.37 to 0.39) and consistent overestimations (average MBE = $0.29 \pm 0.12°C$).

Seasonal variations in forecast performance reveal distinct patterns. For 2-m air temperature, HRRR shows relatively better performance in spring (Fig. S3; RMSE = $2.04 \pm 0.20°C$, $r = 0.90 \pm 0.01$) and fall (Fig. S5; RMSE = $1.81 \pm 0.17°C$, $r = 0.98 \pm 0.01$). Summer forecasts feature a pronounced warm bias across all sites (Fig. S4; average MBE = $0.26°C$), while winter forecasts exhibit a systematic cold bias (Fig. S6; mean MBE = $-0.48°C$). During spring and fall, a clear spatial MBE pattern emerges, resembling the annual pattern of urban cold biases and peripheral warm biases (cf. Fig. 3). Dew point temperature forecasts are least accurate during spring, characterized by a pronounced dry bias (average MBE = $-2.90°C$), higher RMSE (mean = $5.30°C$), and lower correlations (mean $r = 0.86$). The dry bias persists consistently across other seasons as well, albeit to a lesser extent. Wind speed forecasts perform slightly better during winter (mean $r = 0.44$, average MBE = $0.12$ m s$^{-1}$), while overestimations with relatively low correlations are consistent in other seasons.





**Figure 3: Site-specific evaluation of HRRR 18-h forecasts against in situ observations in and around Lubbock, Texas. Rows
correspond to different variables: 2-m air temperature (top), 2-m dew point temperature (middle), and 10-m wind speed (bottom).
Columns show spatial distributions of three statistical metrics: Root Mean Square Error (RMSE; left), correlation coefficient (*r*;
middle), and Mean Bias Error (MBE; right). Urban and rural sites are color-coded by metric value, with urban and rural sites
marked by circles and triangles, respectively.**

In addition to spatial variability, clear forecast performance contrasts emerge between urban and rural sites. As illustrated in
the top row of Fig. 4, rural sites generally show a slight warm bias (average annual MBE = 0.10°C), while urban sites
consistently show a cold bias (average MBE = −0.28°C). This contrast persists across all seasons, with rural sites exhibiting
consistently higher MBE values relative to urban sites. While urban sites show persistent underestimation of 2-m air
temperature throughout all seasons, rural sites exhibit milder cold biases, specifically in fall and winter (average seasonal MBE
= −0.20°C). For dew point temperature, urban–rural differences are minimal, with similar average annual MBEs of −2.00°C





for urban and −1.89 °C for rural sites. Both urban and rural sites consistently show a systematic dry bias across seasons and most pronounced in spring (average seasonal MBE = −2.93°C) compared to other seasons (average MBE = −1.62°C). Wind speed overestimations persistent year-round, with urban sites slightly more biased (average MBE = 0.24 m s$^{-1}$) than rural ones (average MBE = 0.15 m s$^{-1}$).

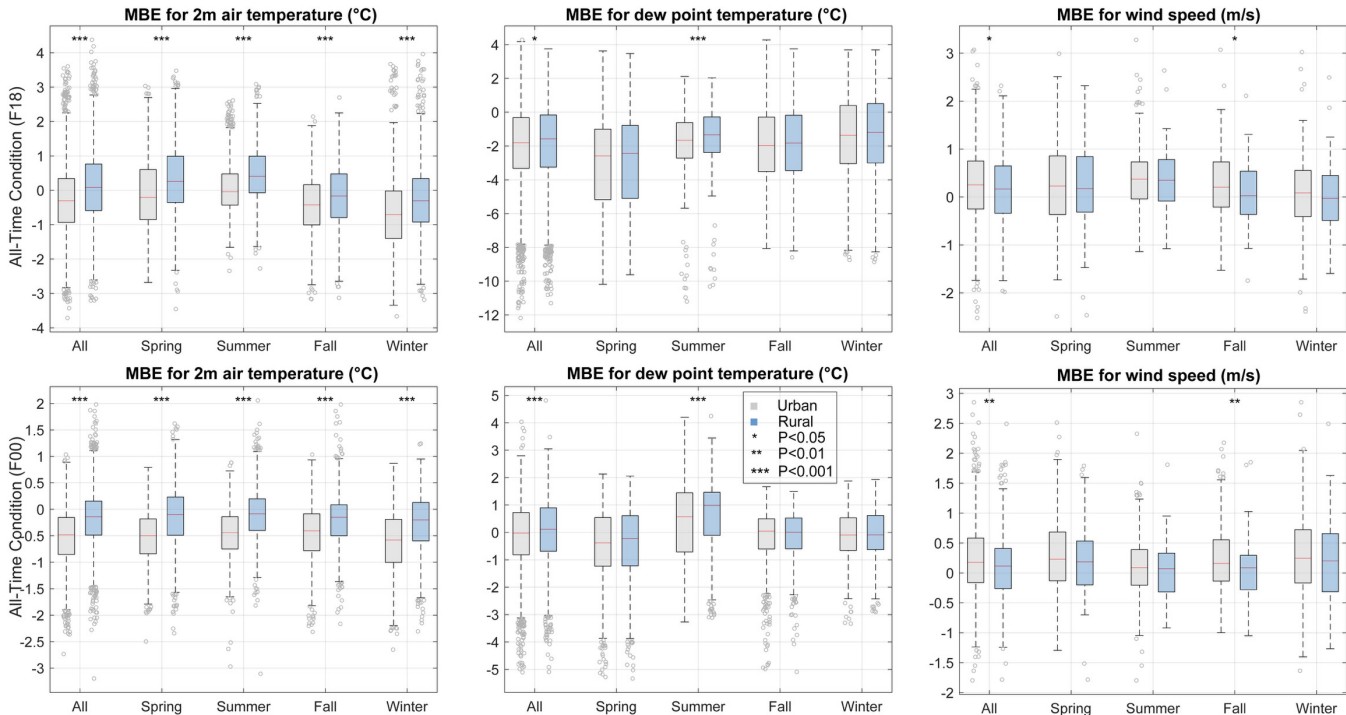

**Figure 4: Diurnal evaluation of HRRR 18-h forecasts using mean bias error (MBE) across all seasons for 2-m air temperature (left), 2-m dew point temperature (middle), and 10-m wind speed (right) based on 18-h lead forecasts (top) and near-real-time (0-h lead) forecasts (bottom). Each pair of boxplots within a seasonal group includes data from all urban and rural site. Each individual data point corresponds to MBE based on a single day at each station location. Box denotes the interquartile range (IQR; 25$^{th}$ to 75$^{th}$ percentile), with the horizontal line indicating the median. Whiskers extend to the furthest data points within ±1.5 × IQR. Asterisks**
**indicate statistically significant differences between urban and rural results based on a paired-sample t-test such that \*, \*\*, and \*\* correspond with p < 0.05, p < 0.01, p < 0.001, respectively.**





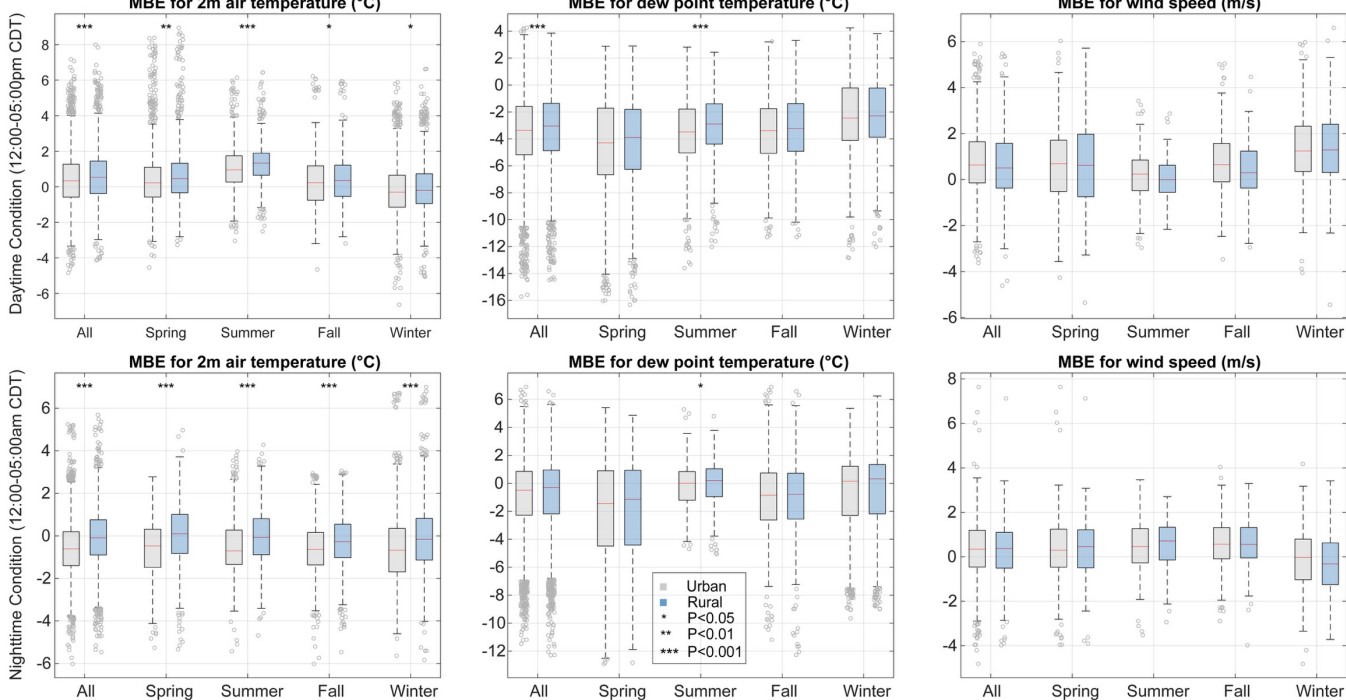

**Figure 5: Temporal evaluation of HRRR forecasts using mean bias error (MBE) across all seasons for 2-**
**m air temperature (left), 2-**
**m dew point temperature (middle), and 10-m wind speed (right) during daytime (12:00-05:00 pm CDT; top) and nighttime (12:00-**
**05:00 am CDT; bottom). Each subplot includes five seasonal groupings (All, Spring, Summer, Fall, and Winter), with paired boxplots**
**for urban and rural sites. Each individual data point corresponds to MBE based on a single daytime or nighttime period at each**
**station location. Box denotes the interquartile range (IQR; 25th to 75th percentile), with the horizontal line indicating the median.**
**Whiskers extend to the furthest data points within ±1.5 × IQR. Asterisks indicate statistically significant differences between urban**
**and rural results based on a paired-sample t-test such that *, **, and ** correspond with p < 0.05, p < 0.01, p < 0.001, respectively.**

We further evaluated HRRR forecasts during daytime and nighttime periods, as shown in Fig. 5. MBEs at both urban and rural

sites are generally larger during the day across all variables and seasons. For 2-m air temperature, HRRR forecasts show a

daytime warm bias at both urban (annual average MBE = 0.38°C) and rural (annual average MBE = 0.59°C) sites. At night,

urban sites exhibit a pronounced cold bias (annual average MBE = −0.57°C), while rural sites show only a slight

underestimation (annual average MBE = −0.09°C). Notably, daytime warm biases are consistent during spring, summer, and

fall, while cold biases dominate during winter. At night, cold biases are present across all seasons, except for a slight warm

bias at rural sites during spring (average MBE = 0.08°C). Additionally, extreme temperature biases (outliers) tend to be less

prevalent during nighttime periods.

For dew point temperature, HRRR forecasts generally yield drier conditions during daytime than nighttime at both urban and

rural sites across all seasons. Spring notably exhibits the strongest dry bias, on average with daytime MBE of −4.59°C (urban)

and −4.28°C (rural), and nighttime MBEs of −1.93°C (urban) and −1.86°C (rural). Dew point forecasts show the best

performance during summer nights, as reflected by narrower MBE spreads and less pronounced negative outliers with mean

MBE of −0.16°C (urban) and 0.01°C (rural). Daytime periods consistently show dry biases across all sites and seasons. Wind



speed forecasts reveal systematic overestimations year-round, with stronger biases at urban sites except during winter daytime.
Although seasonable differences are relatively minor, HRRR forecasts are most accurate during summer daytime and winter nighttime.

**3.2 Comparison of HRRR forecast skill at 0-h and 18-h lead times**

To assess the extent to which forecast performance may deteriorate with lead time, particularly within urban areas, we
compared HRRR model performance between the 18-h lead-time forecasts and the 0-h forecasts, the latter of which incorporates data assimilated on hourly to sub-hourly timescales from multiple observational sources. As shown in the lower row of Fig. 4, the average annual MBEs of 2-m air temperature for the 0-h lead time forecasts (urban: −0.52°C; rural: −0.17°C) do not show marked improvement relative to the 18-h lead time forecast (urban: −0.28°C, rural: 0.10°C). However, the 0-h forecasts display notably lower variability in the temperature bias, with errors generally confined within ±2.5°C, which
indicates improved performance in capturing extreme temperatures.

Across individual seasons, rural sites consistently exhibit smaller MBEs under the 0-h forecasts compared to urban sites, while slightly larger temperature biases are observed using the 18-h forecast. A clear and substantial improvement is observed for dew point temperature in the 0-h forecasts, as suggested by the reduced biases, narrower spreads, and fewer extreme outliers using the 0-h forecasts, both annually and seasonally. Specifically, during summer and winter, average seasonal MBEs are
close to zero, with average MBEs of 0.32°C (urban) and 0.63°C (rural) in summer, and −0.12°C (urban) and −0.08°C (rural) in winter. On an annual basis, average MBE for urban sites markedly improves from −2.0°C (18-h) to −0.11°C (0-h) and similarly for rural sites from −1.89°C to 0.01°C. In contrast, shorter lead time show negligible improvement for wind speed forecasts. The average annual MBEs for wind speed for 0-h forecasts (urban: 0.24 m s$^{-1}$; rural: 0.14 m s$^{-1}$) are nearly identical to those from the 18-hour forecasts, indicating limited sensitivity to forecast lead time.

Overall, the near-real-time forecasts (0-h lead time) generally outperform 18-h forecasts in predicting extreme 2-m air temperatures and dew point temperatures. However, 0-h forecasts do not notably improve wind speed predictions. From an urban–rural contrast perspective, performance gains with shorter lead times are more evident in rural areas, while limited improvements in urban areas may reflect persistent deficiencies in the model's urban representation rather than input data constraints.

**3.3 Nocturnal cooling behavior in HRRR forecasts**

To better evaluate how well HRRR forecasts capture nighttime urban thermal conditions, we compared 18-h forecasted nocturnal cooling rates against in situ observations from both urban and rural sites throughout the study period. Out of 364 complete nights evaluated, 41 exhibited statistically significant nocturnal cooling (p < 0.05), characterized by consistent air temperature decreases at both urban and rural sites (Fig. 6). Among these nights, there was no clear dominance in cooling
behavior between land cover types, as urban sites showed faster cooling on 20 nights, while rural sites cooled faster on 21





nights. Seasonal variations in nocturnal cooling rates were notable, with winter and spring exhibiting the highest cooling rates. Specifically, mean cooling rate during winter was 0.88 °C h⁻¹ at urban sites and 0.93 °C h⁻¹ at rural sites, followed by fall with 0.60 °C h⁻¹ and 0.69 °C h⁻¹, respectively. HRRR forecasts effectively capture urban–rural cooling difference, as rural areas cooled slightly faster. Conversely, summer had the lowest forecasted cooling rates (0.33 °C h⁻¹ at urban and 0.46 °C h⁻¹ at

rural sites), though based on only two clear nights due to frequent cloud cover.

*Rapid cooling events*—defined as cooling rates exceeding 1.00 °C h⁻¹—were forecasted on 27 and 24 nights at urban and rural sites, respectively. These events occurred most frequently during winter (14 urban and 10 rural cases) and fall (13 urban and 12 rural cases). However, no events exceeded 1.50 °C h⁻¹ during the study period. The seasonal distribution of rapid cooling events shows no consistent pattern: based on HRRR forecasts, urban sites recorded one such event in winter and two in spring,

while rural sites recorded three events across fall, winter, and spring. Inter-site variability in cooling rates remained low, as indicated by standard deviations of 0.36 ± 0.01 °C h⁻¹ for both urban and rural sites.





**Figure 6: Evaluation of nocturnal cooling rates on clear nights during the study period. (a) Forecasted cooling rates at urban and rural sites; negative values (red) indicate nighttime warming. (b) Difference between forecasted and observed cooling rates (HRRR - observation); positive values indicate that HRRR predicts stronger (more rapid) cooling than observation. Clear nights are defined as those with domain-wide cloud cover below 25% between 00:00 and 05:00 local time. Only nights with statistically significant cooling rates (p < 0.05) for both HRRR forecasts and observations are shown. Station numbering is listed in Table S1.**

Model performance evaluated using MBE reveals systematic underestimation of nocturnal cooling. At urban sites, the HRRR predicted faster cooling than observations on 15 nights and slower cooling on 29 nights. Rural sites showed a similar pattern, with 13 nights of overestimated cooling rate and 32 nights of underestimation. The average MBE is $-0.12 \pm 0.27$ °C h$^{-1}$ at urban sites and $-0.14 \pm 0.26$ °C h$^{-1}$ at rural sites. Seasonally, except for winter when HRRR exhibited a slight overestimation of cooling rates (urban MBE: 0.05 °C h$^{-1}$; rural MBE: 0.01 °C h$^{-1}$), the model consistently underestimated cooling across all other seasons, averaging $-0.28$ °C h$^{-1}$ at urban sites and $-0.24$ °C h$^{-1}$ at rural sites.

## 3.4 Urban heat advection in HRRR forecasts

Following the application of the filtering criteria outlined in Section 2.6, 110 nights were identified as effective nights suitable for evaluating wind-direction-dependent UHA. Among these, southerly winds, including south, southeast, and southwest directions, dominated, occurring on 62 nights, as shown in Fig. 7. This prevalence may reflect the influence of air masses originating from the Gulf of Mexico.

Observation-based UHA analysis provides a coherent picture of UHA dynamics. Observed temperature anomalies consistently shift downwind under all wind regimes, clearly reflecting the influence of prevailing winds. The observed UHA patterns also reflect the impacts of wind speed; distinct UHA signatures are less evident under both high wind speeds (e.g., northerly winds of 5.35 m s$^{-1}$) and low wind speed (e.g., southeasterly winds of 3.41 m s$^{-1}$). Overall, observations effectively characterize the spatial signature of UHA, revealing clear relationships between wind direction, wind speed, and UHA.

Analysis of UHA patterns derived from 18-hr HRRR forecasts indicates that the model reasonably captures large-scale wind-direction patterns. However, its performance in representing UHA varies considerably among wind regimes. Under southeasterly, westerly, and northwesterly wind regimes, the HRRR effectively predicts the spatial displacement of urban heat downwind of the city core, demonstrating its capability to capture directional UHA patterns. A weak but discernible UHA pattern is also reproduced by the model under southerly wind conditions, suggesting that model captures this feature to some extent. In contrast, under northerly wind conditions, the model incorrectly predicts a cold-core anomaly centered over the urban area. Notably, HRRR-based UHA patterns show limited sensitivity to wind speed, as distinct urban heat advection signatures are not consistently observed across varying wind intensities, except under southeasterly conditions with relatively low wind speeds (i.e., <3.40 m s$^{-1}$).







**Figure 7: Modeled mean Urban Heat Advection (UHA) patterns under eight distinct wind regimes during selected effective nights based on 18-h HRRR forecasts. Black arrows indicate mean wind fields from HRRR forecasts, while red arrows represent observed mean winds from five West Texas Mesonet stations. The gray boundary outlines Lubbock city limits. Colored scatter points depict averaged, normalized temperature anomalies ($T_a$) during each selected effective night.**



## 4 Discussion

Our analysis reveals distinct patterns and complexities in the ability of HRRR forecasts to reproduce near-surface hydrometeorological conditions and urban heat processes within semi-arid urban environments. Across rural sites, HRRR forecasts exhibit a characteristic pattern of overestimated daytime 2-m air temperatures and underestimated nighttime temperatures. This diurnal bias is largely driven by the model's systematic overprediction of incoming shortwave radiation, particularly evident during the warm season (Lee et al., 2023b), also evident in our seasonal analyses (Fig. 4), where summer exhibits higher warm bias at both urban and rural sites than other season. Similar patterns were documented in previous studies for rural areas. For instance, (2019) reported comparable daytime warm and nighttime cold biases at two rural sites in Alabama using 1-h lead time HRRR forecasts, while Lee et al. (2023b) identified a persistent annual warm bias based on 18-h forecasts evaluated against U.S. Climate Reference Network observations. In urban areas, these existing biases are further complicated by HRRR's use of a simplified slab urban model, which omits detailed urban canopy geometry and thus cannot capture the shading and radiative trapping effects of urban street canyons. As a result, despite a positive bias in shortwave radiation leading to higher daytime temperatures (Fig. 5), the nighttime urban environment shows a consistent cold bias due to excessive nocturnal radiative cooling that would otherwise be impeded by urban geometry. The daytime cold bias is presumably driven by enhanced longwave radiation loss, linked to the same shortwave overestimation. This suggests that radiative trapping associated with urban surfaces (e.g., three-dimensional morphology) plays a critical role in shaping near-surface meteorological conditions, underscoring the importance of explicitly representing building structure and shading effects in urban settings.

Regarding dew point temperature, our observations indicate a consistent dry bias across all seasons, especially during the warm season at rural sites. This finding generally aligns with (He et al., 2023), which used 6-h lead time forecasts to evaluate HRRR performance over northern Oklahoma over a full year cycle, an area with climatic and soil conditions analogous to Lubbock. In contrast to (Lee et al., 2019), which did not detect significant dew point biases, our persistent dry bias could be attributed to systematic biases in HRRR's representation of incoming shortwave radiation (positive bias) and precipitation (negative bias) over the southwestern CONUS (Lee et al., 2023b). Interestingly, urban–rural contrasts in dew point temperature biases are less pronounced than anticipated. This diminished contrast likely results from the RUC LSM used in HRRR, known to overestimate soil moisture under dry conditions (soil moisture < 0.2 m³/m³), particularly in semi-arid regions of the southwestern CONUS (Lee et al., 2023b). Similar soil moisture biases have also been found in previous studies using different LSMs (Leeper et al., 2017; Xia et al., 2015). Furthermore, the RUC LSM permits evaporation from urban impervious surfaces modeled as homogeneous slabs, thus maintaining relatively higher soil moisture levels. As a result, this may mask expected urban–rural contrasts in moisture availability.

Wind speed consistently exhibits a positive mean bias throughout all seasons at both urban and rural sites, echoing findings from (Salamanca et al., 2018), which reported similar overestimations using various Noah-MP urban schemes in Phoenix. Our urban sites exhibit a notably larger positive wind speed bias compared to rural areas, likely due to HRRR's slab urban



representation failing to simulate the blocking and drag effects of complex urban structures adequately. However, this contrasts
with recent findings by (Thompson et al., 2025), which evaluated a long-term Noah-MP-based CONUS simulation and
reported a persistent wind speed underestimation at urban sites. This recent study attributed the underestimation to inaccurate
representation of urban roughness lengths and the use of airport weather stations from the CHUWD-H dataset (Wang et al.,
2024) in model evaluations. These contrasting findings highlight the complexity of wind bias patterns and suggest that both

model physics and the choice of observational references play critical roles in determining the reliability of wind predictions
in urban areas.

Comparisons between two forecast lead times indicate that data assimilation, characteristic of near-real-time forecasts, partially
improved HRRR's predictive performance, particularly by reducing extreme values and improving the simulated diurnal cycle.
However, these improvements are not uniformly observed across all variables. Given that accurate surface-layer

parameterizations of heat and moisture exchanges are becoming increasingly critical at longer forecast horizons, incorporating
detailed, up-to-date land surface data through data assimilation remains essential for enhancing HRRR's overall predictive
accuracy. To further investigate the role of land surface characteristics on model performance, we examined relationships
between model biases and fractional land cover within HRRR grid cells. The RUC LSM integrates sub-grid heterogeneity,
which calculates essential surface parameters (e.g., roughness length, emissivity, soil porosity) based on fractional land-use

types instead of the dominant land type (Smirnova et al., 2016). As a result, grids with higher urban fraction and lower
vegetation fraction are expected to exhibit greater model errors. Our analysis confirms this dependency on vegetation and
urban fractions (Figs. S8 and S9). Specifically, increasing vegetation cover leads to reductions in cold biases for 2-m air
temperature in fall and winter and notably decreased dry biases across all seasons except spring, especially during summer, as
it allows enhanced biophysical processes of urban vegetation such as evapotranspiration. Conversely, higher urban fractions

are associated with poorer performance in predicting temperature and dew point. These dependencies underscore the
limitations of HRRR's simplified urban scheme and reinforce the need for more advanced urban canopy models in NWP
systems.

Regarding nocturnal cooling rates, HRRR consistently underestimates the cooling rate relative to observations at both urban
and rural sites. This milder cooling in simulations likely results from modeled excessive soil moisture, incurring slower

nocturnal heat release due to larger soil thermal inertia. Interestingly, the HRRR accurately captures the expected urban–rural
contrast, with urban areas showing slower cooling, which is likely influenced by the interplay between the land surface (high
soil moisture) and atmospheric boundary layer.

Urban heat advection, which is critical to understanding spatial and temporal characteristics of canopy layer UHI, posed
additional modeling challenges. Compared with megacities, small-sized cities are particularly sensitive to dominant synoptic

conditions such as frontal passages (see (Pal et al., 2025) for some example cases over Lubbock region), rendering them more
likely to exhibit weaker urban heat signals. Given HRRR's systematic overprediction of wind speeds at Lubbock, accurately
capturing UHA is inherently challenging, as shown by our results. Yet, previous studies have demonstrated superior
performance of more sophisticated urban schemes, such as those embedded in WRF-urban, which effectively simulate urban



heat related processes such as UHI effects especially under large scale weather impacts (Di Bernardino et al., 2022; Ribeiro et al., 2018). Thus, to advance predictive skill in urban environments, especially in small and semi-arid cities, future operational NWP models may incorporate more realistic urban canopy parameterizations, improve the treatment of land–atmosphere interactions, and account for dynamic surface properties such as detailed vegetation or even anthropogenic heat fluxes.

## 5 Conclusions

We compared HRRRv4 forecasts of key surface hydrometeorological variables, including 2-m air temperature, 2-m dew point temperature, and 10-m wind speed, using in situ observations from 23 densely deployed U-HEAT sites and 5 Mesonet stations in and around Lubbock, Texas. In addition to standard statistical metrics, we also evaluated HRRR's ability to represent urban heat-related processes such as nocturnal cooling and UHA, particularly in the context of a small-sized city located in the semi-arid climate of the southwestern U.S.

During the study period, HRRRv4 demonstrates generally strong agreement with observed 2-m air temperature ($r > 0.95$ and RMSE < 2°C) and dew point temperature ($r > 0.92$ and RMSE < 4°C). However, wind speed forecasts show persistent biases and weaker skill, particularly at urban sites, where the model fails to capture urban drag due to the use of a simplified slab urban model. Characteristic warm bias during the day and cold bias at night are observed at rural sites, while urban sites tend to show mainly cold biases. For dew temperature, a uniform dry bias is observed at both urban and rural sites, especially during the warm season and daytime. Interestingly, the urban–rural contrast in dew point temperature performance is very weak, likely due to overestimated urban soil moisture and evaporation in the model. While HRRR captures the slower nocturnal cooling rates in urban areas than rural surroundings, it generally underestimates cooling rates for both site types. HRRR forecasts also fail to accurately reproduce UHA patterns under most wind regimes due in part to the model's limitations in simulating near-surface wind fields. These deficiencies are likely linked to oversimplified urban parameterizations and inaccurate representation of urban surface characteristics such as thermal inertia, roughness length, and anthropogenic heat emissions.

Our findings highlight the critical importance of appropriate urban representation in NWP models, especially as cities increasingly become focal points for weather-related risks. Although the HRRR incorporates improved MODIS-based land cover information, the current treatment of urban land surfaces is still overly generalized. Future efforts to improve urban hydrometeorological simulations in operational models should prioritize the incorporation of advanced urban canopy parameterization schemes, refined sub-grid land surface heterogeneity, and the assimilation of high-resolution urban observations. More broadly, this evaluation highlights the limitations of applying conventional NWP systems to urban environments without targeted enhancements. As cities face growing challenges from extreme heat and flooding, poor air quality, and evolving land cover, the integration of urban-specific processes into NWP frameworks (Wang et al., 2025) will be essential to ensure accurate, actionable forecasts in both research and operational contexts.



*Code and data availability*. The NOAA Online Weather Data (NOWData) for Lubbock Area are available at https://www.weather.gov/wrh/Climate?wfo=lub (National Weather Service, 2025). Original HRRRv4 outputs are available through https://rapidrefresh.noaa.gov/hrrr/ (National Oceanic and Atmospheric Administration, 2025). The processed

HRRRv4 data, observations from West Texas Mesonet and U-HEAT, and source code are available at https://doi.org/10.5281/zenodo.15885174 (Huang and Wang, 2025).

*Supplement*. The supplement related to this article is available online at [link to be added].

*Author contributions*. YH: Conceptualization, Data curation, Formal analysis, Investigation, Methodology, Software, Visualization, Writing (original draft preparation). CW: Conceptualization, Data curation, Funding acquisition, Investigation,

Methodology, Project administration, Resources, Software, Supervision, Validation, Visualization, Writing (review and editing). TL: Data curation, Investigation, Validation, Writing (review and editing). TD: Data curation, Investigation, Validation, Writing (review and editing). SP: Validation, Writing (review and editing).

*Competing interests*. The contact author has declared that none of the authors has any competing interests.

*Disclaimer*. Publisher's note: Copernicus Publications remains neutral with regard to jurisdictional claims made in the text,

published maps, institutional affiliations, or any other geographical representation in this paper. While Copernicus Publications makes every effort to include appropriate place names, the final responsibility lies with the authors.

*Acknowledgements*. This work was supported by the National Oceanic and Atmospheric Administration (NOAA) under grant No. NA21OAR4590361. CW and YH acknowledge support from the National Aeronautics and Space Administration (NASA) under grant Nos. 80NSSC24K1056 (Early Career Investigator Program in Earth Science), 80NSSC24K0357, and

515 80NSSC25K7496, the U.S. National Science Foundation (NSF) under grant No. OIA-2327435, and the U.S. Geological Survey (USGS) under grant No. G24AC00475. We would like to thank our research partners at the National Wind Institute and the West Texas Mesonet (WTM), particularly Wesley Burgett, Matthew Asel, and Dr. Brian Hirth, for providing the WTM datasets essential to the development of the U-HEAT network. U-HEAT network has been routinely maintained by our community partners in Lubbock NWS, Citizen Climate Lobby and TTU graduate students Diya Das, Hassanpreet Dhaliwal,

and Matthew Hamel. We also would like to thank Dr. Tatiana Smirnova for her help with the retrieval of the HRRR dataset. The scientific results and conclusions, as well as any views or opinions expressed herein, are those of the authors and do not necessarily reflect those of OAR, the Department of Commerce, the South Central Climate Adaptation Science Center, or the USGS. This manuscript is submitted for publication with the understanding that the U.S. Government is authorized to reproduce and distribute reprints for Governmental purposes.

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
