# Peer review of "Urban heat forecasting in small cities: Evaluation of a high-resolution operational numerical weather prediction model"

_EGUsphere, 2025_

## Author Comment (AC1)

**Responses to Executive Editor Dr. Astrid Kerkweg's comment:**

*in the "Code and data availability" section the information how to access the HRRR model code is missing. This information needs to be provided latest upon revision of the article.*

Thanks for pointing this out. We will include details on how to access the HRRR model code, including the specific namelist file, in the revised "Code and data availability" section. The updated section will read:

"**Code and data availability.** The NOAA Online Weather Data (NOWData) for Lubbock Area are available at https://www.weather.gov/wrh/Climate?wfo=lub (National Weather Service, 2025). Original HRRRv4 outputs can be accessed at https://rapidrefresh.noaa.gov/hrrr/ (National Oceanic and Atmospheric Administration, 2025). HRRRv4 uses the Weather Research and Forecasting (WRF) model v3.9.1, which is available at https://www2.mmm.ucar.edu/wrf/users/download/get_source.html (National Center for Atmospheric Research, 2025), with the namelist provided at https://rapidrefresh.noaa.gov/hrrr/wrf.nl.txt (National Oceanic and Atmospheric Administration, 2025). The processed HRRRv4 data, observations from West Texas Mesonet and U-HEAT, and source code are available at https://doi.org/10.5281/zenodo.15885174 (Huang and Wang, 2025)."

References:

National Center for Atmospheric Research: WRF Source Codes and Graphics Software Downloads, https://www2.mmm.ucar.edu/wrf/users/download/get_source.html, last access: 4 October 2025.

National Oceanic and Atmospheric Administration: The High-Resolution Rapid Refresh (HRRR), https://rapidrefresh.noaa.gov/hrrr/, last access: 25 July 2025.

National Weather Service: NOWData - NOAA Online Weather Data, https://www.weather.gov/wrh/Climate?wfo=lub, last access: 25 July 2025.

Huang, Y. and Wang, C.: Evaluation of High-Resolution Rapid Refresh forecasts in small cities. Zenodo [code and data set], https://doi.org/10.5281/zenodo.15885174, 2025.

---

## Author Response (AR1)

**Responses to Reviewer #1's comments:**

*This study comprehensively compared between HRRRv4 forecasts data with situ observations in a small-sized city located in the semi-arid climate of US. The detailed validation conducted on a medium-sized city remote from mega cities is particularly noteworthy, especially the inclusion of verification for urban heat advection. However, the content of this paper, particularly the sections of introduction and discussion, requires revision to highlight the study's key findings and insights.*

Thank you for your valuable feedback. Please see our point-to-point responses below.

*Major comments*

*(1) The study provides a valuable analysis of Lubbock. To broaden the implications, might the authors consider incorporating additional small-sized cities with different climate backgrounds?*

We fully agree that multi-city comparisons would strength the generalizability of urban forecast evaluations as well as scalability of the results presented here. However, conducting a robust evaluation of urban heat forecasts in small cities requires unusually dense, city-scale observational networks to resolve key features such as the urban heat island (UHI) magnitude, nocturnal cooling, and intra-urban temperature/moisture gradients. Small-city networks of comparable density to our Lubbock setup are exceedingly rare. In fact, to our knowledge, the Lubbock network is the only one in the U.S. that provides publicly accessible data at this scale. Extending the analysis to small cities in additional climate regimes would therefore require new observational deployments, which is beyond the scope of this paper and the current project.

In addition, we deliberately chose a semi-arid small city as the initial testbed because drylands are particularly susceptible to climate change (Huang et al., 2017), which may further intensify UHI effects in these regions. Recent global modeling work also indicates that urbanization-driven biophysical warming and associated land–atmosphere feedbacks are stronger in water-limited (dry) regimes than in humid ones, underscoring dryland cities as critical stress tests for operational urban forecasts (Zhang et al., 2025).

To acknowledge this limitation and clarify the study's broader relevance, in the revision, we have (i) clarified in the Introduction section why a dry/semi-arid small city provides a meaningful first testbed, and (ii) added discussion on the transferability of our findings across climates and city sizes, explicitly linking them to known, non-city-specific HRRR performance with appropriate caveats. We have also emphasized the need for future studies to replicate this framework in other small cities where similarly dense observations are available, and called for community efforts to establish and share such networks to enable systematic multi-climate benchmarking. We believe these additions enhance the broader relevance of the paper while remaining aligned with the current scope.

Specifically, we have added the following content in the revised Introduction section (Lines 87–89): "*Recent global modeling work also indicates that urbanization-induced warming and associated land–atmosphere feedback processes are stronger in water-limited (dry) regimes*

*than in humid ones, underscoring dryland cities as critical stress tests for operational urban forecasts (Zhang et al., 2025).*"

We have also added the following content in the revised Conclusions section (Lines 512–520): "*Future work should advance evaluation and model development in parallel. Replicating this analysis in other small U.S. cities that vary in population density, degree of urbanization, land cover, and background climate, with comparably dense within-city observations, will enable more systematic assessments of model performance and provide additional insight into the generalizability and scalability of our results. From a modeling perspective, our findings underscore the need for more realistic urban representations in NWP systems. Future developments should prioritize the integration of advanced urban canopy parameterizations, refined sub-grid land surface heterogeneity, and high-resolution urban observations. More broadly, this evaluation highlights the limitations of applying conventional NWP systems to urban environments without targeted enhancements. As cities face growing challenges from extreme heat and flooding, poor air quality, and evolving land cover, integrating urban-specific processes into NWP frameworks (Wang et al., 2025) and examination of parameterization schemes (Lee et al., 2023a, 2025) will remain essential to ensure accurate, actionable forecasts in both research and operational contexts.*"

**References:**

Huang, J., Li, Y., Fu, C., Chen, F., Fu, Q., Dai, A., Shinoda, M., Ma, Z., Guo, W., Li, Z., Zhang, L., Liu, Y., Yu, H., He, Y., Xie, Y., Guan, X., Ji, M., Lin, L., Wang, S., Yan, H., and Wang, G.: Dryland climate change: Recent progress and challenges, Rev. Geophys., 55, 719–778, https://doi.org/10.1002/2016RG000550, 2017.

Lee, T. R., Pal, S., Krishnan, P., Hirth, B., Heuer, M., Meyers, T. P., Saylor, R. D., and Schroeder, J.: On the Efficacy of Monin–Obukhov and Bulk Richardson Surface-Layer Parameterizations over Drylands, J. Appl. Meteorol. Climatol., 62, 1655–1675, https://doi.org/10.1175/JAMC-D-23-0092.1, 2023a.

Lee, T. R., Pal, S., Meyers, T. P., Krishnan, P., Hirth, B., Heuer, M., Saylor, R. D., Kochendorfer, J., and Schroeder, J.: Impact of the Bowen Ratio on Surface-Layer Parameterizations of Heat, Moisture, and Turbulent Fluxes in Drylands, J. Appl. Meteorol. Climatol., 64, 549–568, https://doi.org/10.1175/JAMC-D-24-0075.1, 2025.

Wang, C., Zhao, Y., Li, Q., Wang, Z., and Fan, J.: Ultrafine‐Resolution Urban Climate Modeling: Resolving Processes Across Scales, J. Adv. Model. Earth Syst., 17, e2025MS005053, https://doi.org/10.1029/2025MS005053, 2025.

Zhang, K., Fang, B., Oleson, K., Zhao, L., He, C., Huang, Q., Liu, Z., Cao, C., and Lee, X.: Urban land expansion amplifies surface warming more in dry climate than in wet climate: A global sensitivity study, J. Geophys. Res. Atmospheres, 130, e2024JD041696, https://doi.org/10.1029/2024JD041696, 2025.

*(2) Given that this study primarily focuses on the evaluation of HRRR forecasts, a more comprehensive overview of previous validation studies concerning HRRR (or other high-resolution operational forecasts) within the introduction would strengthen the argument.*

We appreciate this constructive suggestion. To strengthen our argument, we have added the following summary of previous validation studies to the revised Introduction section (Lines 94–103):

"*The forecast products from HRRR have been evaluated from a variety of perspectives in previous studies. These include assessments of warm-season precipitation over the U.S. Central Plains (Bytheway et al., 2017), cloud cover across the contiguous United States (Griffin et al., 2017), and convective storm characteristics in the eastern United States (Katona et al., 2016). More recent evaluations have focused on convective available potential energy, near-surface meteorology, and surface energy fluxes in Alabama (Lee et al., 2019; Wagner et al., 2019), as well as winds and gusts in New York State (Fovell and Gallagher, 2022). Beyond these evaluations, HRRR forecast products have increasingly been incorporated into urban applications. For example, HRRR forecasts have been coupled with hydrological models to support urban flood forecasting (Coelho et al., 2022) and used to improve air quality predictions (Park et al., 2025). To our knowledge, however, HRRR forecasts have never been systematically evaluated for urban heat dynamics.*"

**References:**

Bytheway, J. L., Kummerow, C. D., and Alexander, C.: A features-based assessment of the evolution of warm season precipitation forecasts from the HRRR model over three years of development. Weather Forecast., 32, 1841–1856, https://doi.org/10.1175/WAF-D-17-0050.1, 2017.

Coelho, G. D. A., Ferreira, C. M., and Kinter Iii, J. L.: Multiscale and multi event evaluation of short-range real-time flood forecasting in large metropolitan areas, J. Hydrol., 612, 128212, https://doi.org/10.1016/j.jhydrol.2022.128212, 2022.

Fovell, R. G. and Gallagher, A.: An evaluation of surface wind and gust forecasts from the high-resolution rapid refresh model. Weather Forecast., 37, 1049–1068, https://doi.org/10.1175/WAF-D-21-0176.1, 2022.

Griffin, S. M., Otkin, J. A., Rozoff, C. M., Sieglaff, J. M., Cronce, L. M., Alexander, C. R., Jensen, T. L., and Wolff, J. K.: Seasonal analysis of cloud objects in the High-Resolution Rapid Refresh (HRRR) model using object-based verification. J. Appl. Meteorol. Climatol., 56, 2317–2334, https://doi.org/10.1175/JAMC-D-17-0004.1, 2017.

Katona, B., Markowski, P., Alexander, C., and Benjamin, S.: The influence of topography on convective storm environments in the eastern United States as deduced from the HRRR. Weather Forecast., 31, 1481–1490, https://doi.org/10.1175/WAF-D-16-0038.1, 2016.

Lee, T. R., Buban, M., Turner, D. D., Meyers, T. P., and Baker, C. B.: Evaluation of the High-Resolution Rapid Refresh (HRRR) Model using near-surface meteorological and flux observations from Northern Alabama, Weather Forecast., 34, 635–663, https://doi.org/10.1175/WAF-D-18-0184.1, 2019.

Park, S., Sayeed, A., Seo, J., Henderson, B. H., Naeger, A. R., and Gupta, P.: Hour by hour PM2.5 mapping using geostationary satellites. ACS ES&T Air, 2, 1816–1830, https://doi.org/10.1021/acsestair.4c00365, 2025.

Wagner, T. J., Klein, P. M., and Turner, D. D.: A new generation of ground-based mobile platforms for active and passive profiling of the boundary layer. Bull. Am. Meteorol. Soc., 100, 137-153, https://doi.org/10.1175/BAMS-D-17-0165.1, 2019.

*At the same time, the reasons for evaluating forecasts rather than reanalysis products should be more clearly presented.*

We interpret the reviewer's use of "reanalysis" as referring to the HRRR 0-h analysis, which represents the model state at forecast initialization. Because HRRR's operational configuration (including model physics and data assimilation) evolves over time, the 0-h analysis should not be treated as a true reanalysis product.

We have clarified in the revised Introduction section why focusing on forecasts, rather than reanalysis products, is critical for operational urban heat applications. Specifically, we have added the following content in the revised Introduction section (Lines 56–60): "*Importantly, NWP forecasts, rather than reanalysis products, are particularly critical for supporting urban resilience and heat mitigation through early warning systems ... Because operational heat warnings depend on lead-time forecast skill, evaluation efforts should focus on forecast fields, whereas reanalysis products, which assimilate observations, may mask systematic model errors and bias performance assessments.*"

We believe that this revision, as well as the revision we made in Lines 94–103, can make our rationale for focusing on forecast skill more explicit.

*(3) Lines 62–77: The introduction of ULSM/UCM is too detailed. Such description may initially lead readers to assume that the paper is about developing or coupling a new UCM into NWP. A more appropriate focus would be on the limited assessment of slab models within operational forecasting.*

Thank you for the helpful suggestion. We agree and have revised this section to better align with our paper's focus. Specifically, we have removed less relevant details regarding the implementation of advanced urban canopy models in various modeling systems, added an explicit statement that this work evaluates the operational HRRR with a slab urban scheme, and condensed this paragraph to emphasize the role of slab models in operational NWP and the lack of forecast-focused evaluations.

The revised paragraph now reads (Lines 64–75):

"*Recent advances in high-resolution urban land-use datasets and urban land surface models (ULSMs) have substantially improved urban representation in numerical models (Chen et al., 2011; Lipson et al., 2024; Stewart et al., 2014). Among ULSMs, one-dimensional slab models remain widely used in operational NWP because they are computationally efficient (Oleson et al., 2008) and perform reasonably well in simulating urban surface energy fluxes over predominantly impervious surfaces with strong sensible heat fluxes (Jongen et al., 2024; Lipson et al., 2024). However, these models idealize the urban surface as a homogeneous layer and oversimplify radiative and hydrological processes that are increasingly important*

in cities with nature-based solutions such as urban vegetation, green infrastructure, and irrigation (Huang et al., 2025; Wang et al., 2025). While slab models have been critisized for their structural simplicity and are often considered surpassed by more advanced urban schemes in research applications, they remain the default choice in many operational forecasting systems. Despite this continued use, there has been limited evaluation of their forecast performance, especially with respect to capturing fine-scale spatiotemporal variations in urban heat. Addressing this gap is crucial for enhancing urban heat forecasting and informing the development of more accurate and adaptive urban land surface parameterizations for operational use."

**References:**

Chen, F., Kusaka, H., Bornstein, R., Ching, J., Grimmond, C. S. B., Grossman‑Clarke, S., Loridan, T., Manning, K. W., Martilli, A., Miao, S., Sailor, D., Salamanca, F. P., Taha, H., Tewari, M., Wang, X., Wyszogrodzki, A. A., and Zhang, C.: The integrated WRF/urban modelling system: development, evaluation, and applications to urban environmental problems, Int. J. Climatol., 31, 273–288, https://doi.org/10.1002/joc.2158, 2011.

Huang, Y., Wang, C., and Wang, Z.-H. Multi-parameterization of hydrological processes in an urban canopy model. Build. Environ., 285, 113567, https://doi.org/10.1016/j.buildenv.2025.113567, 2025.

Jongen, H. J. et al.: The Water Balance Representation in Urban‑PLUMBER Land Surface Models, J. Adv. Model. Earth Syst., 16, e2024MS004231, https://doi.org/10.1029/2024MS004231, 2024.

Lipson, M. J. et al.: Evaluation of 30 urban land surface models in the URBAN‑PLUMBER project: Phase 1 results, Q. J. R. Meteorol. Soc., 150, 126–169, https://doi.org/10.1002/qj.4589, 2024.

Oleson, K. W., Bonan, G. B., Feddema, J., Vertenstein, M., and Grimmond, C. S. B.: An Urban Parameterization for a Global Climate Model. Part I: Formulation and Evaluation for Two Cities, J. Appl. Meteorol. Climatol., 47, 1038–1060, https://doi.org/10.1175/2007JAMC1597.1, 2008.

Stewart, I. D., Oke, T. R., and Krayenhoff, E. S.: Evaluation of the 'local climate zone' scheme using temperature observations and model simulations, Int. J. Climatol., 34, 1062–1080, https://doi.org/10.1002/joc.3746, 2014.

Wang, C., Zhao, Y., Li, Q., Wang, Z.-H., and Fan, J.: Ultrafine‑Resolution Urban Climate Modeling: Resolving Processes Across Scales, J. Adv. Model. Earth Syst., 17, e2025MS005053, https://doi.org/10.1029/2025MS005053, 2025.

*(4) The focus on a small-sized city is an important contribution. To better highlight this value, the results and discussion could both include more explicit comparisons with validation results from large metropolitan areas. For example, how do the prediction errors of HRRRv4 forecasts data found in Lubbock differ (in quantitative terms) from errors reported in studies of large cities?*

Thank you for this valuable suggestion. To our knowledge, quantitative evaluations of urban heat forecasts based on HRRRv4 for large metropolitan areas remain rather limited, and this gap is one of the motivations for our study, albeit conducted in a smaller city. That said, we agree that incorporating quantitative comparisons with previous evaluations would strengthen our discussion. In the revised Discussion section, we have added comparisons with three previous studies, although all of which focus on non-urban sites and/or CONUS-scale evaluations rather than metropolitan areas. Specifically, the following content has been added (Lines 415–419 and Lines 443–450):

"*For instance, Lee et al. (2019) evaluated 1-hour HRRRv2 forecasts of 2-m air temperature using two micrometeorological towers in rural northern Alabama and reported comparable daytime warm biases (average MBE = 0.85 °C) and nighttime cold biases (average MBE = − 0.75 °C). A subsequent evaluation of HRRRv4 against 114 stations of the U.S. Climate Reference Network suggested an average MBE of approximately 0.4 °C for 18-hour forecasts in 2021 (Lee et al., 2023b), with no particular emphasis on dryland cities.*"

"*A recent evaluation of HRRRv4 using 788 Automated Surface Observing System (ASOS) stations across the U.S. found nearly perfect correlations between observed and forecasted 10-m wind speeds, independent of forecast hour or time of day (Fovell and Capps, 2024). However, this evaluation was biased toward well-exposed stations. In the same study, a regional evaluation using 121 New York State Mesonet (NYSM) stations reported an average MBE of 1.22 m s$^{-1}$, which is generally consistent with but slightly higher than our results. Notably, several rooftop urban weather stations in New York City were excluded from this evaluation due to mismatches with model heights (Fovell and Capps, 2024), which further illustrates the current lack of robust urban forecast evaluations.*"

**References:**

Fovell, R. G. and Capps, S. B.: Sustained wind forecasts from the High-Resolution Rapid Refresh model: skill assessment and bias mitigation, Atmosphere, 16, 16, https://doi.org/10.3390/atmos16010016, 2024

Lee, T. R., Buban, M., Turner, D. D., Meyers, T. P., and Baker, C. B.: Evaluation of the High-Resolution Rapid Refresh (HRRR) Model Using Near-Surface Meteorological and Flux Observations from Northern Alabama, Weather Forecast., 34, 635–663, https://doi.org/10.1175/WAF-D-18-0184.1, 2019.

Lee, T. R., Leeper, R. D., Wilson, T., Diamond, H. J., Meyers, T. P., and Turner, D. D.: Using the U.S. Climate Reference Network to Identify Biases in Near- and Subsurface Meteorological Fields in the High-Resolution Rapid Refresh (HRRR) Weather Prediction Model, Weather Forecast., 38, 879–900, https://doi.org/10.1175/WAF-D-22-0213.1, 2023b.

*Minor*

*The acronym urban heat advection (UHA) is only defined in the abstract. Please also spell it out at its first occurrence in the introduction*

Thank you for pointing this out. We have added the full name of UHA at its first occurrence

in the revised Introduction section (Line 108).

**Responses to Reviewer #2's comments:**

*This study evaluates HRRRv4 forecasts against two observational networks in Lubbock, Texas: the dedicated U-HEAT deployed across the city, and the regional West Texas Mesonet. The U-HEAT dataset is a clear strength of the paper and provides a valuable basis and a detailed year-long assessment of systematic model biases. The inclusion of nocturnal cooling rates and urban heat advection in the evaluation is an important contribution as it extends the analysis beyond standard meteorological variables. The manuscript is well organised, with clear sections, and is relevant for both urban climate studies and operational forecasting applications. At the same time, certain aspects of the study could be clarified and extended to further strengthen the generalisability and reproducibility:*

Thank you for your valuable feedback. Please see our point-to-point responses below.

*Major comments:*
*1. Since the study is centred on a single mid-sized city in a semi-arid climate, it would strengthen the conclusions to discuss more explicitly how the identified biases might generalise to other small cities under different climatic conditions. A brief paragraph clarifying transferability across different climatic regimes would help readers gauge generalisability.*

Thank you for your suggestion. We have added a new paragraph to clarify the transferability, and the last two paragraphs of the revised Conclusions now read (Lines 505–520):

"*Although this evaluation focuses on a single small city in a semi-arid climate, several of the identified forecast biases are likely to occur in other small cities under different climatic conditions. This expectation arises primarily from HRRR's use of a slab urban scheme, which simplifies urban surfaces, and is partially supported by previous evaluations of near-surface temperature and wind speed at non-urban sites. However, confirming the transferability of these biases will require dense, city-scale observational networks deployed in additional small cities. This is particularly important because many small urban areas are represented by only a few HRRR urban grid cells, yet can exhibit substantial spatial variability in vegetation fraction, soil moisture, and urban morphology.*

*Future work should advance evaluation and model development in parallel. Replicating this analysis in other small cities with similarly dense within-city observations will enable more systematic assessments of model performance across different climatic regimes. From a modeling perspective, our findings underscore the need for more realistic urban representations in NWP systems. Future developments should prioritize the integration of advanced urban canopy parameterizations, refined sub-grid land surface heterogeneity, and high-resolution urban observations. More broadly, this evaluation highlights the limitations of applying conventional NWP systems to urban environments without targeted enhancements. As cities face growing challenges from extreme heat and flooding, poor air quality, and evolving land cover, integrating urban-specific processes into NWP frameworks (Wang et al., 2025) and examination of parameterization schemes (Lee et al., 2023a, 2025) will remain essential to ensure accurate, actionable forecasts in both research and operational contexts.*"

**Reference:**

Lee, T. R., Pal, S., Krishnan, P., Hirth, B., Heuer, M., Meyers, T. P., Saylor, R. D., and Schroeder, J.: On the Efficacy of Monin–Obukhov and Bulk Richardson Surface-Layer Parameterizations over Drylands, J. Appl. Meteorol. Climatol., 62, 1655–1675, https://doi.org/10.1175/JAMC-D-23-0092.1, 2023a.

Lee, T. R., Pal, S., Meyers, T. P., Krishnan, P., Hirth, B., Heuer, M., Saylor, R. D., Kochendorfer, J., and Schroeder, J.: Impact of the Bowen Ratio on Surface-Layer Parameterizations of Heat, Moisture, and Turbulent Fluxes in Drylands, J. Appl. Meteorol. Climatol., 64, 549–568, https://doi.org/10.1175/JAMC-D-24-0075.1, 2025.

Wang, C., Zhao, Y., Li, Q., Wang, Z., and Fan, J.: Ultrafine‐Resolution Urban Climate Modeling: Resolving Processes Across Scales, J. Adv. Model. Earth Syst., 17, e2025MS005053, https://doi.org/10.1029/2025MS005053, 2025.

*2. The evaluation of nocturnal cooling rates (Sect. 3.3) is informative, but is based on a subset of nights with continuous domain-wide cloud cover below 25% and statistically significant cooling (p<0.05) (Sect. 2.5). To assess robustness, it would help to report how many nights satisfy the cloud-cover filter and to briefly justify the chosen threshold at 25%.*

Thank you for this helpful comment.

Applying both filters, i.e., domain-mean cloud cover below 25% and statistically significant cooling ($p < 0.05$), yields 41 nights. If only the statistical significance criterion is applied, 51 nights are retained. The 25% cloud-cover threshold follows the U.S. National Weather Service definition (https://www.weather.gov/bmx/nwsterms), where 12.5–25% cloud cover corresponds to "mostly clear or mostly sunny" conditions. As a sensitivity test, using a stricter 12.5% cutoff (i.e., clear or sunny) results in 40 nights, with no change in the conclusions.

We have added these counts, the rationale for the threshold, and a note on sensitivity to the revised manuscript (Lines 213–215 and Lines 380–382):

"*... we restrict our analysis to nights with continuous domain-wide cloud cover below 25%. This threshold, corresponding to "mostly clear" conditions in U.S. National Weather Service definitions, is selected to isolate surface-driven cooling processes and minimize cloud-related variability while retaining an adequate sample size.*"

"*Note that we also evaluated the sensitivity of the results to the selection criteria. Using only the statistical significance criterion ($p < 0.05$) yields 51 nights, whereas applying a stricter 12.5% cloud-cover threshold results in 40 nights. The conclusions remain unchanged across these sensitivity tests.*"

*Minor comment:*
*The acronym UHA is introduced in the abstract but not defined at its first occurrence in the main text (Sect. 1, line 97). Please ensure that acronyms are consistently defined when first used in the manuscript.*

Thank you for pointing this out. We have added the full name of UHA at its first occurrence in the revised Introduction section (Line 108).

---

## Author Response (AR2)

**Responses to Editor's comments:**

*Thank you for your thorough revision of manuscript egusphere-2025-3397. You have addressed all reviewer concerns comprehensively, and I am pleased to recommend acceptance for publication in GMD.*

*Before final publication, I have one optional but strongly encouraged suggestion to enhance the paper's value as a self-contained evaluation study. While your revision appropriately condensed the general urban canopy model literature discussion in response to Reviewer #1's feedback, I believe the paper would benefit from a focused technical summary of the specific model being evaluated. I recommend considering an Appendix that provides a concise technical summary of HRRR's slab urban parameterisation scheme, particularly focusing on how the scheme calculates near-surface temperature, specific humidity, and wind diagnostics, along with the key parameterisations for surface energy balance, roughness lengths, and thermal properties, and the physical assumptions and known limitations relevant to urban environments.*

*This enhancement would help readers interpret your findings without needing to consult HRRR technical documentation, which is particularly valuable since your evaluation reveals systematic biases that likely stem from the slab scheme's physical limitations. I suggest approximately one to two pages drawing from NOAA's HRRR technical documentation or relevant WRF physics papers, which would make the paper more useful as a standalone reference for the growing community evaluating operational urban forecasts.*

*This addition is not required for acceptance, and if you prefer to proceed without it, the manuscript is acceptable as-is for immediate publication. Please just reply to me in your next response. However, if you choose to add this enhancement, please submit within two weeks. Either way, congratulations on this excellent contribution to urban NWP evaluation.*

Thank you for this great suggestion! We have now added a new Appendix A to explain "how the scheme calculates near-surface temperature, specific humidity, and wind diagnostics, along with the key parameterizations for surface energy balance, roughness lengths, and thermal properties, and the physical assumptions and known limitations relevant to urban environments". The new appendix is included below for your reference:

**Appendix A: Technical summary of the urban parameterization used by the land surface model in HRRRv4**

This appendix summarizes the slab urban parameterization implemented in the Rapid Update Cycle Land Surface Model (RUC LSM) within HRRRv4. Technical details are primarily based on Dowell et al. (2022) for HRRRv4, Smirnova et al. (2016) for the MODIS-based RUC LSM, and Benjamin et al. (2021) for the diagnostic fields.

*(1) Surface representation and physical properties*

Operational HRRRv4 couples the atmosphere model to the RUC LSM, which includes nine soil layers extending to a depth of 3 m. Urban areas are represented through the "urban and built-up" land-use category in the MODIS classification. The RUC LSM treats each urban

grid cell as a slab surface (i.e., without explicit building geometry), rather than more advanced urban canopy models such as the single-layer urban canopy model.

RUC LSM in HRRRv4 uses a mosaic approach to account for sub-grid land-use heterogeneity. For each grid cell, surface parameters such as emissivity, leaf area index, and plant coefficient for transpiration function are computed as fractional-area weighted averages of all land-use types present. These aggregated values govern the grid-mean properties. However, the effective roughness length is not a simple area-weighted mean. It is computed based on a blending-height formulation following Mason (1988).

Sub-grid soil heterogeneity is similarly represented using the area-weighted mosaic method. Soil hydraulic and thermal properties, including heat capacity, Clapp–Hornberger parameterization exponent, available water capacity, saturated hydraulic conductivity, saturated soil matric potential, residual soil moisture, field capacity, wilting point, and quartz fraction, are averaged over soil types within the grid. The resulting effective parameters are used in the soil heat conduction and moisture transport equations.

In HRRRv4, the greenness fraction has been updated from climatological values to a real-time VIIRS-based green vegetation fraction product, allowing dynamic seasonal evolution of vegetation cover.

*(2) Surface energy balance:*

The RUC LSM solves coupled heat and moisture transfer equations for soil and canopy layers together with a surface energy balance at the interface between the surface and the atmosphere. The net radiation flux at the surface is decomposed into sensible heat flux, latent heat flux, ground (soil) heat flux in the top layer, heat storage, and energy flux of snow phase change. The model solves for surface/skin temperature and specific humidity to close this balance using a root-finding algorithm. Surface exchange coefficients for heat and moisture are from the MYNN surface layer scheme, which provides stability-dependent turbulent transfer coefficients.

*(3) Near-surface temperature, humidity, and wind*

The 2-m air temperature is diagnosed using surface/skin temperature, sensible heat flux, air density, heat transfer coefficient, and potential temperature at the lowest prognostic model level (0.999-sigma or ~8 m above ground level). The 2-m specific humidity is diagnosed from surface and lowest-level specific humidities, latent heat flux, air density, and moisture transfer coefficient. The 2-m dew point temperature is calculated directly from temperature, specific humidity, and pressure at the lowest prognostic model level.

The 10-m winds are estimated by logarithmic interpolation between model levels using Monin–Obukhov similarity, typically between the first and second model levels (0.999-sigma and 0.996-sigma or ~8 m and ~30 m above ground level). The derived 10-m wind represents a grid-cell mean wind, consistent with the grid-mean roughness length.

*(4) Known limitations of the HRRRv4 urban parameterization*

Although HRRRv4 benefits from frequent data assimilation and high spatial resolution (3 km), the representation of urban processes remains simplified. Major limitations include:

**Lack of explicit urban geometry:** Buildings and streets are represented as a uniform slab, with no street-canyon radiative trapping and shading. As a result, nocturnal longwave trapping and diurnal shadowing effects are not captured.

**Prescribed and static urban parameters:** Thermal and radiative properties (e.g., emissivity, roughness, and heat capacity) are fixed for the urban land-use category and do not vary with geographic location, morphology, or building material.

**Simplified surface heterogeneity:** The sub-grid mosaic approach represents fractional contributions to parameters from multiple land use types but cannot resolve within-grid variations and interactions.

**Diagnostic height inconsistency in urban canopy layers:** The 2-m and 10-m output variables essentially assume horizontally homogeneous, aerodynamically smooth surfaces and therefore do not accurately represent conditions within urban canopy layer, where the mean building height often exceeds the first or even second atmospheric model levels.

**Simplified hydrology:** Urban surfaces are not treated as fully impervious; infiltration and evaporation can still occur through soil properties. The model lacks stormwater routing or runoff storage, which limits its direct applications for hydrological or flood studies.

**Omission of anthropogenic fluxes and urban vegetation management:** Anthropogenic heat emissions, building energy use, or irrigation of green spaces, which can modify local energy and moisture balances, are not considered.